# How effective is a blended web-based rehabilitation for improving pain, physical activity, and knee function of patients with knee osteoarthritis? Study protocol for a randomized control trial

**Maria Moutzouri** [ID] *, **Georgios Gioftsos** [ID]

Physiotherapy Department of University of West Attica, Egaleo, Attica, Greece

* Moutzouri@uniwa.gr

## Abstract

Due to the chronic nature of knee osteoarthritis (KOA) self-management is considered an essential part of therapy to improve physical function, activity, pain and quality of life (QoL). Web-based rehabilitation may be a potential innovative mode of patient' training to guide management compared to usual care, especially with the current restrictions pandemic imposed. Moreover, in order to alter KOA patients' behavior towards physical activity (PA), it may be more attractive and motivating to combine within their rehabilitation program, outdoor real life local activity that could feasible to be sustained in the future. Aim of the current study is to evaluate the effects of a blended web-based rehabilitation compared with structured PA alone in patients with KOA. This is a randomized multi-center study with two prospective arms. Fifty-six eligible participants with KOA will be recruited from the West Attica region (considered as structurally weak areas). After a comprehensive face-to face training session, participants will follow a 6-week web-based rehabilitation program, consisting of exercise, advice material enhanced outdoor structured PA. The control group will be encouraged to follow the outdoor structured PA alone. Baseline, 6-week and 12-week follow up assessments will be performed. The primary outcome is self-reported physical function as measured by the Knee Injury Osteoarthritis Outcome Score (KOOS). Secondary measures include pain, function (Timed Up and Go Test, Sit to Stand test), PA levels (Lower Extremity Activity Scale, Baecke Scale and pedometer), psychological perspective (Tampa Scale of Kinesiophobia) and health-related QoL (Short-Form 12). Baseline-adjusted Analysis of Variance will be used to test for group differences in the primary and secondary outcomes. The study will evaluate the blended web-based exercise and advice material, enhanced with outdoor PA in many respects compared to the outdoor PA alone so as to promote self-management care programs for KOA patients.

**Trial registration**: Prospectively registered ISRCTN12950684 (27-09-2020).

relevant data from this study will be made available upon study completion.

**Funding:** YES. This research study is co-financed by Greece and the European Union (European Social Fund- ESF) through the Operational Programme «Human Resources Development, Education and Lifelong Learning» in the context of the project "Reinforcement of Postdoctoral Researchers - 2nd Cycle" (MIS-5033021), implemented by the State Scholarships Foundation (Y). Author MM received the grant, with the number 2019-050-0503-18079. The State Scholarship Foundation had no role in study design, data collection and analysis, decision to publish, or preparation of the manuscript. Its role is soley to fund post-doctoral researchers in novel projects after a blinded process of evaluation of applications.

**Competing interests:** No competing interests.

**Abbreviations:** KOA, knee osteoarthritis; KOOS, Knee Injury Osteoarthritis Outcome Score; Mbq, Modified Baecke questionnaire; MCID, minimal important clinical difference; PA, physical activity; QoL, quality of life; SF-12, short form 12; TSK, Tampa scale for Kinesiophobia; VAS, visual analogue scale; WB-OPA, web based outdoor physical activity.

## Introduction

Knee Osteoarthritis (KOA) is a prevalent, chronic degenerative disease characterised by pain, restricted normal daily activities (ADL) and work absenteeism [1]. Limited activity affects psychological well-being, and quality of life (QoL) [2, 3]. Current care for KOA patients is often not consistent with the clinical guidelines [4–6] recommending strengthening exercises, increase in low impact physical activity (PA) and weight management [7–11]. Alternatively, drug therapy and surgical interventions are over-emphasized and add an economic burden to the Healthcare System [7, 10, 12].

With regards to PA, current guidelines suggest that OA patients should be as physically active as their individual potential and condition allows [13–15]. More specifically, the weekly recommendation for aerobic exercise is 150 minutes of moderate-intensity or 2 days/week of moderate-to-vigorous PA muscle-strengthening exercises [16, 17]. Experiencing benefits from exercise and PA, has shown to shape positive beliefs and motivation towards maintaining activity [18]. Therefore, awareness on self-management, in order to avoid restriction of ADL is essential, as alternatively a vicious circle of inactivity, muscle weakness, weight gain, limited socializing, depression, pain increase and functional impairment has been observed [17, 18].

Despite the positive effects on symptoms [19], exercise-focused programs do not promote sustained behavior change and psychological uplift [20]. Another important limitation of the exercise-focused programs is the low degree of adherence that subjects with chronic OA tend to show [21]. For this reason, other measures i.e. motivational talks, calls are often necessary to improve compliance with these programs. Dynamic facilitators in engaging and sustaining exercise and PA are support by healthcare professionals and community, awareness on the condition, encompassing behavioral interventions, purposeful, enjoyful and flexible modes of PA in chronic pain patients [22]. Web-based rehabilitation could be a more cost-effective option to increase access in structurally weak areas, where appropriate healthcare infrastructure is lacking (healthcare and technology literacy is challenging in addition to poorer social income). Moreover, a great advantage is that it can be performed at a self-determined time, and in a home-based environment, which could enhance adherence and resolve transportation and cost issues. There is mounting evidence that orthopedic technology-assisted rehabilitation has a positive impact on various clinical conditions [23], as well as in KOA [24–27]. More specifically, studies on subjects with KOA that have used a website or an electronic application, as a support, for teaching exercise have shown positive effects for increasing PA [24–30].

Given the above, there is a need for a blended program that encompasses physical and behavioral elements, with purposeful and flexible PA, as well as interplay of self-management and healthcare support strategies to promote a good fit for the care of KOA. The study primarily aims to compare the efficacy of a 6-week web-based rehabilitation program of exercise, advice enhanced with outdoor structured PA to manage pain and physical function in KOA patients compared to outdoor PA program alone; secondarily maintenance of the outcomes at mid-term (12-weeks follow up period) will be examined.

## Hypotheses

The proposed randomized controlled trial (RCT) will test the following three hypotheses: 1) A 6-week blended web-based rehabilitation program will be more efficacious in improving PA levels than a 6-week enhanced outdoor activity program alone immediately following the intervention. 2) A 6-week blended web-based rehabilitation program will be more efficacious in improving pain and self-reported physical function than a 6-week enhanced outdoor activity program alone following the intervention. 3) A 6-week blended web-based rehabilitation

program will be more efficacious in improving psychological function, QoL and physical activity levels than a 6-week enhanced outdoor activity program alone at mid-term (12 weeks).

## Materials and methods

### Trial design

This will be an assessor-blinded, parallel group, 2-arm prospective randomized controlled trial (Fig 1). The protocol will conform to SPIRIT guidelines for reporting randomized controlled trial studies (S1 Checklist). The trial has been prospectively registered in the ISRCTN clinical trial registry (ISRCTN12950684/27-09-2020, https://www.isrctn.com/ISRCTN12950684). Ethics approval was granted by the Ethics Committee of the University of West Attica, Greece (49238/09-07-2020).

A sample of 56 patients with painful KOA will be recruited from the community of West Attica, Greece. The selection of these municipalities was based on the intention to support weak, relevantly neglected infrastructure and poorer financially resources this region holds. A number of recruitment strategies will be used including (i) advertising through then municipality, community centers, local newspapers, Peristeri KEP Ygeias and University websites, University staff newsletters, and Facebook; (ii) placing brochures and flyers and study posters in medical and physiotherapy clinics; (iii) conducting presentations about knee OA in the local community.

### Participants

**Eligibility criteria.** i. Aged ≥ 45 years; ii. Diagnosis of KOA (Kellgren Lawrence ≥ Grade 1–3); iii. Knee pain for ≥ 3 months; iv. Reporting average knee pain in the last week ≥ 3 on an 11-point Numeric Pain Rating Scale (NPRS); v. ability to write and speak Greek; xi. able to use/access computer or tablet and have access to the internet.

**Exclusion criteria.** Knee surgery within the past 6 months; ii. Planning any back or lower limb surgery within the next 12 months; iii. Current or past (within 3 months) oral or intra-articular corticosteroid use; iv. Systemic arthritic conditions such as rheumatoid arthritis or gout; v. Physiotherapy, chiropractic or acupuncture treatment specifically for the knee within the past 6 months; vi. Inability to walk unaided (assistive device such as cane or walker can be

| | Enrollment | Allocation | Post-allocation | | Close -out |
|---|---|---|---|---|---|
| | | | **Study period** | | |
| **Timepoint** | -t1 | Baseline (0) | 0-6weeks | 6-12weeks | 12week endpoint |
| **Enrollment:** | | | | | |
| | | | | | |
| Eligibility screening | x | | | | |
| Informed Consent | x | | | | |
| Randomization | x | | | | |
| Allocation | | x | | | |
| **Interventions** | | | | | |
| Intervention group: WB-OPA | | | ←——————→ | | |
| Usual care group: OPA | | | ←——————→ | | |
| Both groups: self-managed OPA | | | | ←——————→ | |
| **Assessments** | | | | | |
| Baseline | | x | | | |
| 6 weeks | | | x | | |
| 12 weeks | | | | | x |

**Fig 1. Schedule of enrollment intervention and assessments.** WB-OPA: web-based outdoor physical activity; OPA: outdoor physical activity.

used) as this is necessary for some of the physical testing; vii. Medical condition precluding safe exercise (i.e. uncontrolled hypertension or heart condition); viii. Psychiatric history or cognitive impairment precluding safe compliance to the program; ix. Neurological condition.

## Interventions

During the first face-to-face session (week 1), the physiotherapist (MM) will provide information about KOA and the importance of PA. This introductory session will be scheduled for all participants (both groups) to get familiarized with the webpage environment. Moreover, the journey route of the outdoor walking program, appropriate for each participant will be selected and mutually agreed for both groups. To ensure consistency in content and delivery, the same physiotherapist (with >16 years of postgraduate clinical experience) will take care of the supervision and progression of the program of all participants. Weekly phone supervision to encourage adherence and resolve any issues i.e. potential overload or motivation will be offered to both groups. The weekly dose will be completed for 6 weeks in the community. The participants will be instructed to walk at a moderate level of intensity determined by the Rate of Perceived Exertion Scale (0–10). After completion of the 6-week program (week 7), participants will be encouraged to continue improving PA levels in a second face-to-face session, especially with the walking journey routes, but will not receive any additional intervention till the final follow-up. A final face-to-face appointment will take place in week 12 to support and encourage patients to maintain a PA lifestyle.

**Blended web-based and outdoor PA program (WB-OPA).** Participants randomized to blended web-based rehabilitation (based on ESCAPE-knee pain online resources) will participate in an exercise-based rehabilitation program designed to improve physical function by integrating exercise, PA, education on the condition, and self-management strategies. ESCAPE-pain stands for Enabling Self-management and Coping with Arthritic Pain using Exercise. It is an evidence-based, cost effective, rehabilitation program for people with chronic OA that combines a progressive exercise program with simple advice strategies on how to self-manage their condition [25, 26]. ESCAPE-pain delivers the NICE core recommendations of exercise and education for the management of OA. The physiotherapist teaching the exercise program to participants has undertaken the accredited ESCAPE-pain course. The blended character of the rehabilitation is based on two components: a) physical exercise delivered via web-based video and outdoor PA, and b) education management strategies delivered again via web-based video. In the introductory face-to face session, individual passwords will be provided to this group in order to register and have access to the rehabilitation program. Participants will be instructed to attend 12 exercise sessions, for 6 weeks, twice weekly. More specifically, participants will perform a 35–40 minutes simple exercise regimen based on pre-recorded video that will progress as they improve. The exercises train elements of joint flexibility, muscle strength, balance, and muscle endurance. Additionally, each week, participants will be encouraged to attend a 5–10 minutes pre-recorded advisory video session by the physiotherapist. The advisory sessions include information on OA related topics, i.e. risk factors, clinical manifestations, feasible strategies like ice, dietary control, pacing, goal setting, and overcoming psychological distress. Table 1 presents the study schedule.

In addition to the web-based exercise program, the experimental group will be prescribed to perform thrice weekly a pre-determined walk of 500-800m (a walking dose of 70 minutes per week in separate sessions of at least 10 minutes duration), according to their individual potential. The journey routes selected, based on the distance, safety, comfort and enjoyable paths (i.e. local green parks, pedestrian shopping malls, sports courts) with a relevant map will be provided (with analytic description, photos of key corners, benches for rest etc.). Walking

**Table 1. Description of the study schedule.**

| Baseline | • Eligibility criteria assessed by investigator |
|---|---|
| | • Read patient information letter |
| | • Sign informed consent |
| | • Completing baseline assessment measurement |
| | • Schedule face- to face appointment for week 1 |
| Introductory face-to -face session | • Provision of information on OA, benefits of PA on pain by PT |
| week 1 | • Familiarization patient with webpage environment |
| | • Registration patient and provide digital codes for entrance in the exercise and education video-material |
| | • Instruction of exercises of week 1 in the home environment |
| | • Selection and agreement on outdoor journey route, provide relevant map |
| | • Instruction of the use of pedometer |
| | • Instruction of the use/ completion of diary weekly log |
| | • Determination of goals |
| | • Video-education session 1 |
| | • Two video home-exercise sessions |
| | • Schedule phone call for week 2 |
| Week 2- week 5 | • Video-education session 2–5 |
| | • Two video home-exercise sessions |
| | • Completion of log-book |
| | • Discussion of progress by phone call supervision with PT |
| | • Schedule phone call for following weeks |
| | • Schedule appointment for follow-up session of week 6 |
| Week 6 | • Video-education session 6 |
| | • Two video home-exercise sessions of week 6 |
| | • Completion of log-book |
| | • Follow-up assessment measurement week 6 by investigator |
| | • Face-to-face session with PT to monitor progress, logbook and goals on exercise and outdoor activity, discuss increase of selected outdoor PA |
| | • Encouragement of continuing exercise and PA |
| | • Encouragement to ask phone call supervision (at their own will) |
| | • Schedule a follow-up assessment of week 12 |
| Week 7–11 | • Self-selected home exercise |
| | • Selected outdoor activity (walking route) |
| | • Schedule appointment for week 12 |
| Week 12 | • Final follow up assessment by investigator |
| | • Face-to-face session with PT to discuss long-term goal and support for maintaining active lifestyle. |

programs are known to have beneficial effects on knee pain and function for people with mild to moderate KOA [27, 31–36].

**Usual care.** Participants randomized to usual care (pragmatic control arm) will be guided to engage PA in their daily routine. In the first session, they will be introduced to the same webpage, where the general information on KOA will be only available for education purposes. Participants will be also guided in this session to select a journey route (mutually agreed) within the community and following the same instructions and parameters as the experimental group with the only difference of a frequency of 5 times weekly. The reason for the

increased frequency in this group (5 times per week versus 3 times per week in the experimental group) is to preserve an equal dosage of exercise between groups, (since the experimental group will follow the exercise program twice weekly).

Participants in both groups will continue taking their usual medications and other non-surgical treatments to manage their KOA, and use normal assistive devices such as a cane if needed.

**Adherence strategies.** To increase the likelihood of adherence to the interventions, the following behavioural change techniques and strategies will be used. First, each participant will have a planning session with a physiotherapist of up to 30 min to plan the location, day and time of day for each walk, and reinforce that each walk was moderate intensity in at least a 10 min. Second, regular physiotherapy supervision and monitoring each week with regular phone calls based on patient preference will be offered. If any complication or query should arise, participants could discuss it in the weekly phone calls or contact directly the physiotherapist, and they will be informed of the procedure to follow. Third, each participant will wear a pedometer and record the number of steps taken and time spent walking during each session in a logbook. All participants will complete the daily logbook to record the time (in minutes) spent in activity either walking or exercising in for each walking session over the 12 weeks (Table 2). The physiotherapist will monitor the participant's logbook at each weekly supervisory session (by phone). Fourth, participants will be encouraged to engage social supports such as walking with a friend, family member if they chose to. They will also describe any changes to their usual care of their KOA and any problems with their knee while doing their walking program each week.

Participants in both groups will be free to withdraw from the study at any time, but the withdrawal rates and reasons will be recorded.

## Outcomes

**Primary outcome measures.** Outcome measures have been selected based on those recommended for clinical trials of OA [37–39].

a). Physical function: the Knee Injury Osteoarthritis Outcome Score (KOOS) Likert version, which is a disease-specific instrument with good psychometric properties demonstrated in range of OA studies [38, 40]. The KOOS includes 42 items in 5 separately scored subscales: Pain, Other Symptoms, ADL, Function in Sport and Recreation (Sport/Rec), and Knee related QoL.

b). Pain: NPRS week Average knee pain over the previous week measured by a valid and reliable Numerical Pain Rating Scale pain with terminal descriptors 0 (no pain) and 10

**Table 2. Logbook diary to monitor compliance, duration and frequency of exercise and outdoor walking activity weekly.**

|  | Week 1 | | Week 2 | | Week 3 | | Week 4. . . | ..Week 10 | Week 11 | Week 12 |
|---|---|---|---|---|---|---|---|---|---|---|
|  | Exercise (-/min) | Walking (-/min/step) | Exercise (-/min) | Walking (-/min/ steps) | Exercise (-/min) | Walking (-/min/ steps) | Exercise (-/min) | . . . | . . . | Walking (-/min/ steps) |
| Monday |  |  |  |  |  |  |  |  |  |  |
| Tuesday |  |  |  |  |  |  |  |  |  |  |
| Wednesday |  |  |  |  |  |  |  |  |  |  |
| Thursday |  |  |  |  |  |  |  |  |  |  |
| Friday |  |  |  |  |  |  |  |  |  |  |
| Saturday |  |  |  |  |  |  |  |  |  |  |
| Sunday |  |  |  |  |  |  |  |  |  |  |

(worst possible pain and has been shown to be preferred by patients with chronic pain NPRS over other measures of pain intensity, including the VAS, due to comprehensibility and ease of completion [41, 42].

**Secondary outcome measures.**

c). Physical Function:

- The 30-second Chair Stand test (30 CTS) provides a direct, objective measure (number/ times participants can rise to a full standing position from sitting and return to sitting, with arms crossed and held against the chest, in 30 seconds will be counted). The 30 CTS has been recommended within the minimal core set of performance-based outcome measures in OA research and clinical practice [38, 43].

- The timed up and go (TUG) test evaluates walking speed and mobility and has been found as valid and reliable in KOA patients [43, 44]. Participants are instructed to stand up from a standard height chair and walk at their normal pace around a marker 3 meters away before returning to the chair and sitting down again. TUG has been found as valid and reliable in KOA patients [45].

d). PA level:
  Habitual PA will be measured in three ways, 1) using a questionnaire, 2) second using a scale and 3) using a pedometer.

- The Modified Baecke Physical Activity Questionnaire (mBQ), validated in samples adults ≥ 55 years of age [46, 47], to assess PA.

- The UCLA scale indicates patients' most appropriate activity level, with 1 defined as "no PA, dependent on others" and 10 defined as "regular participation in impact sports", and has been found as the only scale discriminating between insufficiently and sufficiently active patients in KOA [48].

- A pedometer (Yamax SW200 digi-walker) has been compared with actual steps in multiple studies [49–54] and consistently performs well. This pedometer will be clipped to the participants' waistband on either the left or right hip, each morning and will be removed it at bedtime for 3 consecutive days [54] on three occasions (baseline, week 6 and week 12) to record the number of steps taken per day.

e). Health-related QoL: This will be assessed using the Short-Form-12 QoL instrument which is widely used measure of general health and QoL status [55].

f). Psychological Function: Tampa Scale for Kinesiophobia (TSK). The TSK will be examined to provide a brief fear of movement scale that is valid and reliable in people with OA [56, 57].

## Sample size

A clinically meaningful difference for the physical function measure of KOOS is considered to be 15%. The margin was derived from a minimal important difference KOOS-pain subscale (MCID$_{80}$ KOOS-PS) score of 10 units reported for the KOOS based on the study by Lyman [58] in KOA patients. It was estimated that individual randomization would require 22 patients in each group participants per arm for a trial with 80% power to detect a 15% difference between trial arms, with a 5% significance level (2-tailed). Fifty-six participants will be recruited to take a 20% drop out rate estimation into account.

**Randomisation- Allocation concealment.** Recruited participants will be randomly allocated to either the experimental group or control group at a 1:1 ratio. Random numbers will be generated using a computer software program run by an external statistician. After baseline assessment, participants will be provided an envelope according to the randomization sequence by a volunteer undergraduate who will prepare consecutively numbered, sealed, opaque envelopes. The envelopes will be kept in a locked location accessible only by the unblinded physiotherapist (MM). The physiotherapist will then schedule the participants' first appointment.

The investigator (AK) performing all assessments will be blinded to the participant randomization assignment and will not be involved in providing the interventions. Participants will be requested not to disclose details about their treatment to the outcome investigator. The statistician will be blind to group allocation until completion of the statistical analyses.

**Statistical analysis.** We will calculate the proportion of enrolled participants out of all eligible patients, the adherence rate and the dropout rates. The Shapiro-Wilk Test will be used to determine the normality of the data. Descriptive statistics will be presented for each group as mean change (standard deviation (SD), 95% confidence intervals) for all outcomes from baseline to 12 weeks. Analysis of covariance (ANCOVA) will be performed using IBM SPSS version 24 for all the primary and secondary outcome measures with baseline measures as covariates. In the case of missing data in the relevant directions according on how to handle missing data in every outcome measure will be followed.

## Discussion

Evidence so far suggests that strengthening exercises offers significant improvements in pain and function [24, 58], but is not associated with reductions in depression [59]; thus, education for pain coping and cognitive behavioral skills improves psychological functioning in KOA patients [22, 59–64]. This study aims not only to investigate the benefits of blended web-based rehabilitation (exercise and coping education skills implemented in a self-managed manner with minimal supervision) and enhanced with outdoor PA, but also whether these potential benefits can help patients change their level of PA, behavior and perceptions related to KOA, and therefore their routine daily life.

### Strengths and limitations

The study design has several strengths. First, considerable attention has been paid to quality control with the contribution of the established award-winning program ESCAPE-knee pain. Online ESCAPE-knee pain has been found to improve functioning, understanding, and confidence by combining education, advice, and simple coping strategies with functional exercise material [25, 26]. The program does not require specialized training, sophisticated exercises, or equipment for the participants, so it could be implemented and replicated easily. Second, the outcome measures selected are valid, responsive and reliable, cover a range of clinically important concepts, and include those recommended for clinical trials of OA [62]. Third, a well-designed web-based intervention in which patients' can report their experiences with home exercises via phone calls, provide physiotherapists information about patients' individual needs for guidance. Research in web-based interventions has focused on interventions without human physical support. Unfortunately, the effects of these interventions have been shown minimal, especially in the long-term [65, 66]. These minimal effects may be partly explained by the absence of face-to-face guidance [67]. Therefore, the combination of web-based intervention with face-to face minimal supervision as well as the inclusion of real life outdoor activity is more promising [65]. The study has been designed with attention to

methodological quality characteristics such as randomization, concealed allocation and blinded outcome assessment. Finally, the novelty of the study is that the blended program will be provided in weak structured areas where cost-effective options of physiotherapy care are mostly needed.

Limitation of the study is the unblinded nature of the physiotherapist delivering intervention and guidance. In the case of accessibility limitations during the assessment sessions due to the restrictions imposed by COVID-19 measures will be dealt with telecommunication. Moreover, the level of PA will not be measured objectively prior to the beginning of exercise program.

## Supporting information

**S1 Checklist. SPIRIT checklist for trials.**
(DOCX)

**S1 File. Translation of study protocol submitted to IKY 18079.**
(PDF)

## Acknowledgments

We express our gratitude and acknowledge the support and input from Prof Michael Hurley and Andrea Carter of the ESCAPE pain program. We thank the study subjects for their participation and the West Attica municipalities for their willingness to incorporate the current program to their citizens.

No previous presentation of the research in any form.

## Author Contributions

**Conceptualization:** Maria Moutzouri.

**Formal analysis:** Maria Moutzouri.

**Funding acquisition:** Maria Moutzouri.

**Investigation:** Maria Moutzouri.

**Methodology:** Maria Moutzouri.

**Project administration:** Georgios Gioftsos.

**Supervision:** Georgios Gioftsos.

**Validation:** Maria Moutzouri.

**Writing – original draft:** Maria Moutzouri.

**Writing – review & editing:** Georgios Gioftsos.

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
