## [Decision Letter · Decision Letter 0]

15 Mar 2022

PONE-D-21-28366

How effective is a blended web-based rehabilitation for improving pain, physical activity, and knee function of patients with knee osteoarthritis?”

PLOS ONE

Dear Dr. Moutzouri,

Thank you for submitting your manuscript to PLOS ONE. After careful consideration, we feel that it has merit but does not fully meet PLOS ONE’s publication criteria as it currently stands. Therefore, we invite you to submit a revised version of the manuscript that addresses the points raised during the review process.

You may find the comments of one of the reviewers appended to this email. In addition, since this is a study protocol with no published results, this should be stated in the title. 

We look forward to receiving your revised manuscript.

Kind regards,

Jose María Blasco, Ph.D.

Academic Editor

PLOS ONE

“No competing interests.”

5. Please ensure that you include a title page within your main document. You should list all authors and all affiliations as per our author instructions and clearly indicate the corresponding author.

6. Please amend your authorship list in your manuscript file to include author’s names.

7. Please amend your manuscript to include your abstract after the title page.

8. Your ethics statement should only appear in the Methods section of your manuscript. If your ethics statement is written in any section besides the Methods, please delete it from any other section.

10. Please include your tables as part of your main manuscript and remove the individual files. Please note that supplementary tables (should remain/ be uploaded) as separate ""supporting information"" files.

11. "We noticed you have some minor occurrence of overlapping text with the following previous publication(s), which needs to be addressed:

- https://www.oarsijournal.com/article/S1063-4584(16)30486-1/fulltext

- https://viewer.joomag.com/paramedics-paramedics-health-promotion-program/0389408001492717768?page=60

- https://link.springer.com/article/10.1186/1471-2474-13-129?code=e67f92a2-7839-42a8-823d-b3cf43ce3416&error=cookies_not_supported

- https://www.nivel.nl/sites/default/files/bestanden/e-Exercise_proefschrift_Corelien_Kloek.pdf

- https://onlinelibrary.wiley.com/doi/abs/10.1002/acr.20642

In your revision ensure you cite all your sources (including your own works), and quote or rephrase any duplicated text outside the methods section. Further consideration is dependent on these concerns being addressed.

Reviewers' comments:

Reviewer's Responses to Questions

**Comments to the Author**

1. Does the manuscript provide a valid rationale for the proposed study, with clearly identified and justified research questions?

Reviewer #1: Yes

2. Is the protocol technically sound and planned in a manner that will lead to a meaningful outcome and allow testing the stated hypotheses?

Reviewer #1: Yes

3. Is the methodology feasible and described in sufficient detail to allow the work to be replicable?

Reviewer #1: No

4. Have the authors described where all data underlying the findings will be made available when the study is complete?

Reviewer #1: Yes

5. Is the manuscript presented in an intelligible fashion and written in standard English?

Reviewer #1: Yes

6. Review Comments to the Author

You may also provide optional suggestions and comments to authors that they might find helpful in planning their study.

Reviewer #1: Thanks for the opportunity given to me to review the article tittled: “How effective is a blended web-based rehabilitation for improving pain, physical activity, and knee function of patients with knee osteoarthritis?” Please, find the comments on the attached document

7. PLOS authors have the option to publish the peer review history of their article (what does this mean?). If published, this will include your full peer review and any attached files.

Reviewer #1: **Yes: **Fernando Dominguez-Navarro

---

## [Author Response · Author response to Decision Letter 0]

3 Apr 2022

Dear Editor,

Re: submission of an original article for consideration for publication

Please find enclosed a manuscript for consideration for publication in the Medicine for the titled: “How effective is a blended web-based rehabilitation for improving pain, physical activity, and knee function of patients with knee osteoarthritis?” Study protocol for a randomized control trial.

Thank you for taking the time to review our manuscript and for all constructive feedback offered. We have considered all comments made by the Academic Editor and the Reviewer and we believe they have all helped the manuscript to move forward.

Regarding the comments made by the Academic Editor

1.We have ensured that your manuscript meets PLOS ONE's style requirements, including those for file naming. 

2. We have corrected the information regarding grant numbers for the awards we received for our study in the ‘Funding Information’ section.

3. We have added “No competing interests.” in our cover letter. Thank you for changing the online submission form on our behalf.

4. Regarding the repository information if the manuscript is accepted for publication, we have not understood exactly what was exactly expected. Any data asked by the Academic Editor or Reviewers can be offered upon request. However, no storage file is needed and the since the manuscript is a study protocol the data file is now being processed. If you still consider that to be essential please let us know how we can proceed to create the repository information. We have ticked the relevant box saying that we will need help for the creating a DOI. Please update our Data Availability statement to reflect this information.

5.We have included a title page within our main document and have listed all authors and all affiliations as per our author instructions.

6. We have amended our authorship list in our manuscript file to include author’s names.

7. We have amended our manuscript to include our abstract after the title page.

8. We have deleted the Ethics information from any other section apart from the Methods.

9. We have included captions for our Supporting Information files at the end of our manuscript and have updated in-text citations to match accordingly. 

10. We have included our tables as part of our main manuscript and removed the individual files. 

11. We have shown caution and addressed to the minor occurrence of overlapping text with other previous publication(s). We apologise for that. In our revision we have ensured proper citation of all sources, and quoted or rephrased any duplicated text outside the methods section. 

Regarding the specific comments made by the Reviewer:

Introduction

Line 20. Another important limitation of the exercise-focused programs is the low degree of adherence that subjects with KOA tend to show. For this reason, other measures (motivational talks, calls, etc.) are often necessary to improve compliance with these programs.

We agree with comment and have added the statement in the paragraph.

Line 26. There are examples of studies carried out on subjects with KOA that have used a website or an electronic application as a support for teaching physical exercise. It would be appropriate for this paragraph of the introduction to include some reference to these studies, as well as note the possible effects it may have on subjects with KOA por increasing PA. In this way, this paragraph could be more targeted to the condition being studied.

We have added the relevant references in the section.

Material and methods

Line 52: What does it mean "exploratory" in this context? Is this concept necessary?

You are right it is not necesseary in this context since relevat research has been made up to a point, so we have removed the term.

Line 81: If the level of physical activity performed by each subject prior to the start of the intervention has not been assessed, this should appear as a limitation, or at least be mentioned in the discussion.

The level of physical activity will be assessed at baseline with the Baecke scale and UCLA. However, will not be assessed objectively, so we have added it as a limitation in the discussion, lines 290-1.

Line 95: You mention that both groups receive the introductory session, but please also specify that the control phone calls are also received by both groups.

We have added the observation for the phone calls.

Line 104: Please, add the reference

The reference has been added.

Line 126: During the introduction, the need for a blended intervention of physical activity and behavioral programs has been mentioned. However, when it comes to explaining the intervention of the experimental group, this structure is not so clear. I suggest that the authors partially rewrite this paragraph in such a way that the blended structure and how each of the components is worked on is clearer, so that the use of the term "blended" can be justified.

The paragraph has been partly re-written to clarify the blended component (lines 106-128).

Usual care paragraph: To my understand, the walking journey that perform subjects in usual care group is also mutually agreed during the initial session, as explained before. Am I right?

However, it is not clear if they receive any explanation on how to make the walking journey, with what parameters and what factors were used to establish these characteristics, as was detailed in the experimental group. If they simply receive the advice to walk, please specify.

Any justification of why control group performed journey route 5 times per week and the experimental only 3 times?

Yes you are right, the same parameters and characteristics were explained for both groups. The observation has been added to offer the clarification. The frequency was more (5 times per week versus 3 times per week in the experiment group) in the usual care group so as to sustain the dosage equal between groups. In this way both groups will have activity/ exercise for 5 times per week.

Lines 154-155: It is already explained. Please, remove this sentence.

Thank you, it has been removed.

Discussion

Lines 172-176: If pain was already assessed by the specific domain of the KOOS scale, is there any compelling reason to assess it with the NPRS week Average knee pain as well? Please, explain if it the case.

NPRS has been shown to have high reliability, specifically in elderly and less educated patients, and is useful for the assessment of chronic pain. Moreover, it can be used as a quick marker for patients to easily understand and monitor their progress whereas KOOS is more complex and needs more sophisticated scoring. Pleas let us know if you feel we need to add a referenced explanation in the manuscript. 

Lines 245-249: I also understand that one of the objectives is whether this type of intervention contributes to increasing physical activity levels in people with KOA.

Yes, thank you for the observation. We have added the relevant objective.

Please feel free to contact me by telephone or via email, if you have any questions regarding the manuscript. I look forward to hearing for your reply.

Yours sincerely,

Maria Moutzouri

---

## [Editor Report · Decision Letter 1]

5 May 2022

How effective is a blended web-based rehabilitation for improving pain, physical activity, and knee function of patients with knee osteoarthritis? Study protocol for a randomised controll trial.

PONE-D-21-28366R1

Dear Dr. Moutzouri,

We’re pleased to inform you that your manuscript has been judged scientifically suitable for publication and will be formally accepted for publication once it meets all outstanding technical requirements.

Kind regards,

Jose María Blasco, Ph.D.

Academic Editor

PLOS ONE
---

## [Editor Report · Acceptance letter]

12 May 2022

PONE-D-21-28366R1 

How effective is a blended web-based rehabilitation for improving pain, physical activity, and knee function of patients with knee osteoarthritis? Study protocol for a randomized control trial 

Dear Dr. Moutzouri:

I'm pleased to inform you that your manuscript has been deemed suitable for publication in PLOS ONE. Congratulations! Your manuscript is now with our production department. 

Kind regards, 

on behalf of

Dr. Jose María Blasco 

Academic Editor

PLOS ONE